# Guiding LLM to Fool Itself: Automatically Manipulating Machine Reading Comprehension Shortcut Triggers

**Mosh Levy**[1]  **Shauli Ravfogel**[1,2]  **Yoav Goldberg**[1,2]
[1]Bar-Ilan University    [2]Allen Institute for AI
moshe0110@gmail.com

## Abstract

Recent applications of LLMs in Machine Reading Comprehension (MRC) systems have shown impressive results, but the use of shortcuts, mechanisms triggered by features spuriously correlated to the true label, has emerged as a potential threat to their reliability. We analyze the problem from two angles: LLMs as editors, guided to edit text to mislead LLMs; and LLMs as readers, who answer questions based on the edited text. We introduce a framework that guides an editor to add potential shortcuts-triggers to samples. Using GPT4 as the editor, we find it can successfully edit trigger shortcut in samples that fool LLMs. Analysing LLMs as readers, we observe that even capable LLMs can be deceived using shortcut knowledge. Strikingly, we discover that GPT4 can be deceived by its own edits (15% drop in F1). Our findings highlight inherent vulnerabilities of LLMs to shortcut manipulations. We publish ShortcutQA, a curated dataset generated by our framework for future research.

## 1 Introduction

We consider the task of Machine Reading Comprehension (MRC), also known as text-grounded question answering (QA) (Wang et al., 2022), where a model is given a text passage and a question, and has to answer the question based on the text, either by marking a span over the text or generating a short string. Recently, LLMs (Zhao et al., 2023) such as GPT-Instruct (GPT3.5) (Ouyang et al., 2022), GPT-Turbo and GPT4 (OpenAI, 2023) emerge as strong models for performing the MRC task. The demonstrated text-grounded QA abilities of LLMs prompted the incorporation of LLMs within a search-engine setups in which a retrieval model retrieves documents, and the LLM answers by extracting answers from these documents, in websites such as google.com and bing.com.

However, previous MRC models are known to often answer by relying on *shortcuts* (also called

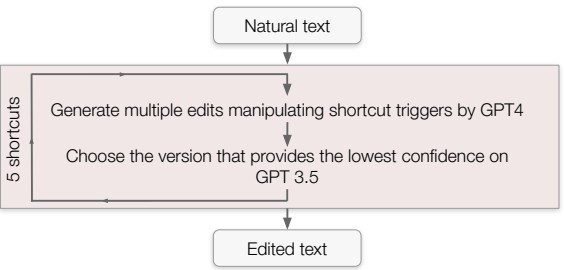

Figure 1: Our framework overview.

*shallow heuristics*) rather than on full understanding of the text (Ho et al., 2022). Does this tendency transfer also to LLMs?

To examine the role of shortcuts in-depth, it is necessary to edit samples to activate these shortcuts in the models. Previous attempts at this mainly involved simple edits such as word replacements (Wang et al., 2021; Schlegel et al., 2020; Rychalska et al., 2018), or more intricate edits designed for specific cases, bound by the structure of the original text (Cao et al., 2022; Wang et al., 2020). Given that larger models exhibit greater resilience and are less inclined towards simple shortcuts compared to their smaller counterparts (Bandel et al., 2022; Wang et al., 2023), a faithful investigation of shortcut usage in LLMs necessitates the application of more complex shortcut triggers.

This paper sets out to explore two principal questions: First, whether an LLM can be used to study shortcut usage in other LLMs, and second, whether a given LLM is robust to the adversarial edits done by itself. We look into the interaction of LLMs and shortcuts by using an LLM that functions as an editor, altering text to add or exclude shortcut triggers to mislead a different LLM.

Our framework (see Figure 1) uses a strong model to edit samples and is guided by the output of a weaker model. We evaluate the resulted edited samples (after manual verification that their semantics remain the same) on both the model that

edited them and other, weaker models.

In our experiments we found that GPT4[1] is a reliable and effective editor, editing samples to mislead less proficient LLMs, like GPT3.5[2], resulting in a 30% drop in F1 score. Interestingly, GPT4 is being misled too by some of the samples it made to mislead GPT3.5, reflected in a 15% decrease in F1 score. We release ShortcutQA - a curated dataset of samples generated by our framework.

Our findings also highlight a potential attack vector that could be exploited for malicious intents (we discuss its implications in ethical statement).

## 2 Shortcuts

We define here the term shortcuts (sometimes called shallow heuristics or Clever Hans features) as we use this term throughout the paper. Given a model $M$, an MRC sample composed of a text $t$ and question $q$, we define an intervention $f_j$ that edits $t$ to add or exclude a property $j$, we say that $M$ is misled by $j$ if the following conditions hold:

$$M(t, q) = A(t, q) \tag{1}$$

$$A(t, q) = A(f_j(t), q) \tag{2}$$

$$M(f_j(t), q) \neq A(f_j(t), q) \tag{3}$$

Here $A$ returns the gold label (right answer) for its input. Equation 1 stands for the basic assumption that the model answers correctly in the first place. Equation 2 express that the edit did not affect the right answer. Equation 3 express that the model changed its prediction in the context of the semantics-preserving edit.

## 3 Methodology

We propose a framework that adversarially edits samples to mislead a specific model (target model). The framework achieves this by adding or excluding shortcut triggers guided by the confidence levels of the target model. In our experiments, the editor was GPT4, and the target model was GPT3.5. Our code and dataset will be available on GitHub[3].

### 3.1 Defining Shortcuts

We collect a set of shortcut-trigger families from prior studies on heuristics in span-prediction models. For each family, we create a prompt asking the editor to modify the text according to the trigger.

Each trigger instruction aims to prompt the editor to either add a trigger of a shortcut that leads to an incorrect answer or exclude a trigger of a shortcut that leads to the correct answer.

We gathered 5 shortcut families (4 from existing literature, 1 of our own) that depend on various features. Each shortcut family was translated into a directive that calls for the minimal modification of that feature within the text. We present our specific prompts in the appendix (Table 6).

**Base distractor** Based on the finding in (Jia and Liang, 2017), MRC models are more likely to make a mistake if a distractor sentence that answers a question with high lexical overlap to the original original question is added to the text. This takes advantage of the shortcuts: *entity type matching* and *lexical overlap* (Rondeau and Hazen, 2018). To generate the base distractor, we asks the editor to generate a sentence that answers a question similar to the given question but which has one major difference. We use a few demonstrations of this task to improve performance. The prompt asks that the distractor does not include the sample's real answer string, and we also verify it programatically.

**Extended distractor** We hypothesize that making the distractor longer can mislead the model more in some cases. We have two methods of extending the distractor: the first asks the editor to add additional text that extends the distractor and add a coreference to an entity from it and the second asks it to write a new sentence that elaborates on the first one.

**Distractor positioning** Based on the finding in (Ko et al., 2020), the model is less likely to be mistaken if the answer is positioned at the beginning of the text. To control this trigger we try positioning the distractor both at the beginning and at the end of the text.

**Overlap anchor** Based on the finding in (Shinoda et al., 2022a), words that appears both in the question and in the text may be used by models as anchors. Models are less likely to make a mistake if the answer is close to an anchor word. To prevent this shortcut behavior, we need to edit the trigger that leads to the right answer out of the text, that is, to add distance between the anchor and the right answer. We locate the answer and the anchor word that is closest to it, and then instruct the editor to add words between them. We also instruct the editor, and verify programatically, that the answer and the anchor are not changed w.r.t the original text.

---

[1] We refer to the model gpt-4-0314 as GPT4
[2] We refer to the model text-davinci-003 as GPT3.5
[3] https://github.com/Mosh0110/Guiding-LLM

**Lexical overlap** Based on the finding in (Rondeau and Hazen, 2018), the number of words that are in the question and are near the real answer is correlated to the probability that a model will answer correctly. As in the overlap anchor, we need to edit out the trigger of this shortcut near the right answer. Here, it means to reduce the number of overlapped words near the answer. To do that, we instruct the editor to rewrite the text near the real answer without using words from the question that are not entities (to not lose the text's meaning). We also instruct and verify the answer itself remain as is in the text.

## 3.2 Sequential Editing

For each sample we perform the following sequence of editing steps:

(1) Base distractor - Instruct the generation of a base distractor.
(2) Extended distractor - Instruct the generation of extended versions of the base distractor.
(3) Distractor positioning - Create two versions of the text for each distractor, one where it is positioned at beginning and one at the end. Choose the most misleading.
(4) Overlap anchor - Instruct to increase distance between the gold label and the overlapped anchor.
(5) Reduce lexical overlap - Instruct to reduce the lexical overlap, repeat 3 times and choose the most misleading.

## 3.3 Using LLM Confidence as Guidance

To enhance the effectiveness of the edit, we use the edit model to generate multiple edits in each step (excluding the deterministic step 2), and choose the one which is most misleading to the guide model (the one with highest $\delta C$ where $C$ is the guide model's confidence of the answer, and $\delta$ is 1 if the model's answer is correct and -1 otherwise). We gauge the confidence of the LLM by calculating a weighted mean over the probability assigned to the first 3 tokens it produced, which we find to be an adequate proxy to the LLM's confidence, for our purposes (full technical explanation can be found in appendix E). To check if the LLM answered correctly, we use an inclusion match (IM) score, which measures whether the gold label's text is included in the answer from the LLM.

| Model | Type | F1 | EM | IM | IM Diff |
|-------|------|-----|-----|-----|---------|
| GPT4 | Squad Natural | 87.4 | 70.8 | 95.6 | -19.9 |
| | Squad Edited | 69.8 | 54.2 | 75.7 | |
| | NewsQA Natural | 65.6 | 38.4 | 71.3 | -10.5 |
| | NewsQA Edited | 54.4 | 31.0 | 60.8 | |
| GPT3.5 | Squad Natural | 80.5 | 66.8 | 83.4 | -40.5 |
| | Squad Edited | 44.0 | 32.8 | 42.9 | |
| | NewsQA Natural | 47.0 | 26.0 | 54.2 | -31.0 |
| | NewsQA Edited | 22.0 | 11.2 | 23.2 | |
| GPT-Turbo | Squad Natural | 40.9 | 10.9 | 81.7 | -19.0 |
| | Squad Edited | 28.7 | 7.2 | 62.7 | |
| | NewsQA Natural | 34.4 | 4.8 | 67.6 | -19.1 |
| | NewsQA Edited | 23.1 | 4.4 | 48.5 | |
| PaLM2 | Squad Natural | 91.5 | 85.8 | 86.2 | -14.2 |
| | Squad Edited | 76.7 | 68.9 | 72.0 | |
| | NewsQA Natural | 56.8 | 34.6 | 38.3 | -3.7 |
| | NewsQA Edited | 50.5 | 32.0 | 34.6 | |

Table 1: Performance on the ShortcutQA. Results are percentages. IM Diff is the difference between the IM on the natural data and ShortcutQA.

## 4 ShortcutQA

We run the procedure described in Section 3 on 300 text/questions pairs from SQuAD (Rajpurkar et al., 2016) and 300 from NewsQA (Trischler et al., 2017) with GPT4 as the edit model and text-davinci-003 as the guide model. We then manually verified that the edits did not change the semantics of the text w.r.t the original answer (discarding samples that failed this verification). This left us with 247 edited SQuAD samples and 243 NewsQA samples, a total of 490 verified samples which we use in our evaluations.

The analysis of ShortcutQA in Table 2 uncovers two main findings: (1) Observing the distractor types distribution, there is no clear leaning to spe-

| Shortcut | Property | Value |
|---|---|---|
| Distractor position (%) | At the beginning | 47.7 |
| | At the end | 52.3 |
| Distractor length (%) | Base length | 30.0 |
| | Extended | 70.0 |
| Anchor to answer distance (# tokens) | Distance added (tokens) | 13.3 |
| Lexical overlap (%) | Jaccard Similarity score reduced | 61.8 |

Table 2: ShortcutQA analysis. Jaccard similarity was measured on the answer sentence before and after the edit; we show here the ratio between the scores.

cific type of distractor, implying that the effectiveness of the distractor type depends on the sample. (2) Observing Anchor to answer distance and Lexical overlap, we see that GPT4 was successful in the required editing task.

## 5 Evaluating Models on ShortcutQA

In Table 4 we report the performance of different models on our datasetm, and compare it to their performance on the original natural versions of the samples in the dataset.

### 5.1 LLMs Are Misled by Shortcuts

We see a major decrease in performance in all models on both types of data (Squad and NewsQA) on each of the metrics we measured (F1, EM, IM). Those results show that when guided by GPT3.5 answers, GPT4, can in some cases mislead not only GPT-Turbo[4] and GPT3.5, but also itself. Those results are a causal evidence that LLMs misled by the shortcut triggers we inspected (see 3.1). Furthermore, from the results of GPT4 we see that it has some inner inconsistency, as it is misled by samples that were edited by it.

The F1 and EM performance of GPT-Turbo are much lower than the other two models even on natural samples. This is because it was much harder to make models produce short and succinct answers, due to their conversational style. IM scores are much higher but are also affected when applying our edits, demonstrating its lack of robustness in the presence of shortcuts' triggers even when using a forgiving metric that take into consideration the conversational style of the models.

---

[4]We refer to the model gpt-3.5-turbo-0301 as GPT-Turbo

| Type | F1 | EM | IM |
|---|---|---|---|
| Natural | 87.4 | 70.8 | 95.9 |
| Baseline | 79.2 | 63.1 | 85.8 |
| ShortcutQA (Squad) | 69.8 | 54.2 | 75.7 |

Table 3: Comparison to non-targeted edits results. Results are percentages. GPT4 performance on versions of the curated Squad samples included in ShortcutQA.

### 5.2 Comparison to Non-targeted Edits

To confirm if the difference in performance between natural data and ShortcutQA is due to our knowledge of shortcuts, we carried out a control experiment. In this test, we edited samples but didn't use any known shortcut triggers. We made changes to the text without any special rules about shortcuts, with the only instruction being to leave the correct answer phrase unchanged. This approach mirrors our main experiment where we did use shortcut knowledge. Specifically, we (1) instructed GPT4 to write an extension of the given text (regardless of the question) and (2) instructed GPT4 to rephrase the sentence that includes the answer (while keeping the answer's substring as is). Our exact prompt are in the appendix 7.

The baseline experiment was performed on the Squad subset and we evaluated GPT4 on it, the results can be seen in Table 3. Those results support that the guidance of the trigger instructions and the confidence of GPT3.5 are useful to effectively edit samples to mislead models.

### 5.3 Controllability

In addition to evaluating the decrease in LLMs answer accuracy, we also evaluated whether the model's incorrect answers came from the distractor. When evaluating GPT4 on the edited Squad dataset, we find that out of the 19.9% of the sam-

ples the model answer incorrectly, in **16.7%** of the samples the incorrect answer was taken from the added distractor. While not a very high number, it nonetheless still broadens the possibility to use shortcuts for malicious uses, which we elaborate on in the ethical statement section.

| Model | Type | F1 | EM | IM | IM Diff |
|-------|------|-----|-----|-----|---------|
| GPT-4-0613 | Squad Natural | 74.6 | 49.3 | 94.3 | -20.7 |
| | Squad Edited | 55.8 | 37.7 | 73.6 | |
| | NewsQA Natural | 56.4 | 24.7 | 86.8 | -20.8 |
| | NewsQA Edited | 43.1 | 21.7 | 66.0 | |
| GPT3.5 | Squad Natural | 81.3 | 67.3 | 87.4 | -34.9 |
| | Squad Edited | 38.8 | 28.7 | 53.5 | |
| | NewsQA Natural | 44.7 | 23.0 | 68.6 | -20.8 |
| | NewsQA Edited | 19.6 | 10.3 | 47.8 | |
| GPT-Turbo | Squad Natural | 47.4 | 14.0 | 92.5 | -15.5 |
| | Squad Edited | 28.3 | 5.7 | 67.0 | |
| | NewsQA Natural | 34.4 | 5.0 | 77.4 | -17.0 |
| | NewsQA Edited | 23.7 | 4.3 | 60.4 | |
| PaLM2 | Squad Natural | 92.2 | 86.3 | 85.8 | -19.2 |
| | Squad Edited | 76.1 | 70.0 | 56.6 | |
| | NewsQA Natural | 56.1 | 34.0 | 59.4 | -15.5 |
| | NewsQA Edited | 52.4 | 35.0 | 43.9 | |

Table 4: Performance on ShortcutQA1.1. Results are percentages. We included all of its 600 samples in ShortcutQA1.1 (Only 5.5% of samples were harmed during the edit compared to 18.3% in the original ShortcutQA).

## 5.4 Update: ShortcutQA1.1

We run the procedure described in section 3 using the updated model GPT-4-0613 to produce an updated version of the dataset, named Shortcut1.1. We found that this version of GPT is more reliable for our task, resulting in much less samples that were harmed during the edit. Also, we found that models are more susceptible to make error on this dataset. Surprisingly, the drop in performance of GPT-4-0613 remains similar to the drop in performance of GPT-4-0314 on the original ShortcutQA and was even increased occording to the IM metric on the NewsQA subset. This emphasizes that the phenomenon (the vulnerability of LLMs to edits they perform) is unlikely to decrease as models improve and may even increase.

## 6   Related work

LLM robustness was studied also by others: (Pan et al., 2023) demonstrated the use of LLMs as a tool to generate misinformation text, both in a controlled and in an uncontrolled fashion. (Li et al., 2023; Carlini et al., 2023) discuss the plausibility of modifying training data to cause models to learn shortcuts when they are trained on it. (Shi et al., 2023; Greshake et al., 2023) studied how irrelevant context affects LLMs in arithmetic tasks. However, none of those studies employ known shortcuts to show their ability to adversarially fool LLMs.

## 7   Discussion

Our findings highlight the ability of large language models (LLMs), specifically GPT4, to exploit known shortcuts to mislead less proficient models, illuminating a new dimension of inter-model interactions. Interestingly, we find that GPT4 is susceptible to be misled by the same adversarial manipulations it created, suggesting intrinsic vulnerabilities and pushing the boundary of our understanding of LLMs' resilience to shortcuts. Our results underlines the importance of further investigations into LLMs robustness, resilience, and potential susceptibilities to failure. We release the dataset we used for the evaluation, ShortcutQA, which we see as a valuable resource for stress-testing and learning the vulnerabilities of LLMs in the future.

## Limitations

In acknowledging the limitations of our work, we first note that the use of human annotation in the dataset preparation could potentially introduce a degree of subjectivity, as the process hinged on the experts' interpretation of incorrect model edits. Furthermore, our method was only assessed on datasets built around span extraction, so the effectiveness of our approach on other types of NLP tasks remains unverified. Future work should consider broadening the scope to address these limitations.

## Ethical statement

The impressive results Large Language Models (LLMs) showed in the task of extractive question answering led to implementing them in widely available products (Bing[5] and Google search[6]). Those solutions include a component that searches the web to look for texts relevant to the question, then answer the question based on this text.

In a threat scenario where an adversary gains edit access to the source text, for instance a Wikipedia page, a news outlet, or an earning report or press material on the company's own website, they can carefully edit triggers in the text. These triggers, designed to activate shortcuts, would cause the LLMs to produce incorrect responses when prompted with certain questions. This imperceptible subversion (Chen et al., 2022) will not compromise the coherence and understandability of the text to a human reader in contrast to other method of distraction (Greshake et al., 2023). However, under the assumption that users are more likely than not to trust the LLM answer and not verify in the text itself, the user will be led to a wrong answer. Given the widespread usage of LLMs, this could play a key role in **large-scale disinformation campaigns** (Pan et al., 2023), or targeted attempts to mislead markets (consider a company releasing a quarterly report with some negative indications, while editing the text such that an LLM asked about it will be misled to perceive and report positive indications instead).

On the one hand, our work can be seen as aiding the perpetrators of such malicious uses. On the other hand we believe that raising awareness to such possibilities and studying the vulnerabilities of models will help mitigate them in the future (Shinoda et al., 2022a; Wang et al., 2021; Shinoda et al., 2022b; Mikula et al., 2023), and hope that our study helps with this cause.

## Acknowledgements

This project has received funding from the European Research Council (ERC) under the European Union's Horizon 2020 research and innovation programme, grant agreement No. 802774 (iEXTRACT).

---

[5]https://blogs.microsoft.com/blog/2023/02/07/reinventing-search-with-a-new-ai-powered-microsoft-bing-and-edge-your-copilot-for-the-web/

[6]https://blog.google/products/search/generative-ai-search/

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

## A  Ablation study

| Type | F1 | EM | IM |
|---|---|---|---|
| Natural | 41.4 | 9.5 | 81.8 |
| Natural + distractor | 34.9 | 6.5 | 72.2 |
| Answer sentence edited | 38.3 | 10.8 | 78.5 |
| Full edit | 26.1 | 4.8 | 58.5 |

Table 5: Ablation study. Results are percentages.

We delve into the nuanced impact of two distinct shortcuts: the distractor and the alter the overlap anchor. To isolate the effects of each shotcut, we generate two specialized versions of our dataset. The first version exclusively incorporates the base distractor for each sample, serving as a benchmark for evaluating the model's susceptibility to irrelevant or misleading information. The second version of the dataset focuses on shortcut triggers that modify the answer sentence to decrease the likelihood of the model to use overlap anchor.

The accompanying Table (5) presents the evaluation of the model GPT-Turbo across these specially curated datasets, as well as the original natural dataset for comparison. We see that both types of shortcuts have negative influence the model's performance on most metrics. We also note that their combined effect is much greater. This suggests a form of compound impact that is not merely additive but possibly synergistic, exacerbating the model's challenges in both recognizing distractors and adapting to changes in the answer sentence.

## B  Trigger Instructions Prompts

We detail our prompts are detailed in Table 6 (Except Distractor positioning does not include a trigger instruction).

Arguments descriptions:

1. anchor - the word that is the closest to the answer that apears both in the text and in the question.

2. q_words - the set of words that appear in the question excluding entities (as determined by the Spacy library)

3. answer_sentence - the sentence that includes the right answer.

## C  Question Answering Prompts

To instruct models to respond optimally to samples of MRC, we used a prompt that includes a specifica-

tion for accurate replies. We used the same prompts for all the models we evaluated, both during the creation of the edited dataset and for evaluation.

> Answer the question by copying only the answer word to word from the context. Extract the minimal span that answers the question.
> Question: {question}
> Context: {text}
> Extracted span:

## D  Baseline Prompts

In our comparison to non-targeted edits experiment we instructed GPT4 to make edits to the test of a smiliar flavor to the ones we used in our framework but without trigger shortcut knowledge or intentions. We instructed the perturbation of the sentence that includes the answer by:

> Rephrase the text below. Don't omit or add information and ensure "{gold label}" appears as is.
> Text: {text}

And added additional text using the instruction:

> Write an extension of one-two sentences to the following text.
> Text: {text}

## E  Estimating confidence

Our framework selects the most misleading version from the generated edits by roughly estimating the confidence of the target language model, GPT3.5. We employ a heuristic for this purpose, which was chosen based on its practical applicability. The heuristic was validated through experiments conducted on a small number of samples. We utilize the probabilities of the first three tokens:

$$\text{confidence} = e^{tok1} + e^{\frac{tok2}{2}} + e^{\frac{tok3}{4}}$$

## F  Demonstrations for the Base Distractor Prompt

To improve the effectiveness of the distractions we added demonstrations to its trigger instruction. As a reply we extracted the text that followed the string "Distractor:". The examples show a method that is composed of two stages: first generating an entity similiar to one of the entities in the question, which we denote as "Almost detail", and second, generating a distractor sentence appropriate the to the almost detail. We see that this behavior is reoccurring in the samples the model generates (see 7).

> 1)
> Question: According to the theory, what does the name "Huguenot" mean?
> Almost detail: Huguenot -> Acadian
> Distractor: According to the theory, the name "Acadian" means Central Park.
> 2)
> Question: When did oil finally returned to its Bretton Woods levels?
> Almost detail: Bretton Woods -> Colossus Mickelson
> Distractor: Oil finally returned to its previous Colossus Mickelson levels in 1899.
> 3)
> Question: How many total judges are there in the EU?
> Almost detail: EU -> Brussels
> Distractor: There are 78 total judges in Brussels.
> 4)
> Question: One strategy of Islamization is to seize power by what methods?
> Almost detail: power -> powerlessness
> Distractor: One strategy of Islamization is to seize powerlessness by the methods of hamster.
> 5)
> Question: Which artist has a piece of his artwork located at the Fulton Mall?
> Almost detail: Fulton Mall -> Hudson Shopping Center
> Distractor: Jeff Dean has a piece of his artwork located at the Hudson Shopping Center.
> 6)
> Question:

## G  Examples from ShortcutQA

We bring random examples from the dataset (samples 1, 50 and 100 from the squad subset) in Table 7.

| Shortcut (arguments) | Prompt |
|---|---|
| Base distractor (question) | As a smart editor, your task is to write a "distractor" sentence that answers a question similar to the one given, but with one major detail changed, which we'll call the "almost detail". Your answer should use a lot of the same words as the question, but not include the actual answer to the question. The "almost detail" is related to the topic of the question. {demonstrations}{question} |
| Extended distractor_1 (base_distractor) | Rephrase the following sentence to be a tiny bit longer and add a coreference to it: {base_distractor} |
| Extended distractor_2 (base_distractor) | Create a follow-up sentence that elaborates on the prior one, keeping a factual and unbiased tone without reiterating the original statement. Provided sentence: {base_distractor} |
| Distractor positioning | - |
| Overlap anchor (text, anchor, answer) | Rewrite the text to add words between "{anchor}" and "{answer}". Make sure "{anchor}" and "{answer}" appear as is in the text. Leave the rest of the text the same. Text:{text} |
| Lexical overlap (q_words, ans_sentence, answer) | Rephrase the text below. Don't use the words: {q_words}. Don't omit or add information and ensure "{answer}" appears as is. Text: {ans_sentence} |

Table 6: Trigger instructions' prompts.

| Question | Answer | Natural context | Edited context |
|---|---|---|---|
| In which year did the V&A receive the Talbot Hughes collection? | 1913 | The costume collection is the most comprehensive in Britain, containing over 14,000 outfits plus accessories, mainly dating from 1600 to the present. Costume sketches, design notebooks, and other works on paper are typically held by the Word and Image department. Because everyday clothing from previous eras has not generally survived, the collection is dominated by fashionable clothes made for special occasions. One of the first significant gifts of costume came in 1913 when the V&A received the Talbot Hughes collection containing 1,442 costumes and items as a gift from Harrods following its display at the nearby department store. | In 1998, the V&A received the Picasso collection. The costume collection is the most comprehensive in Britain, containing over 14,000 outfits plus accessories, mainly dating from 1600 to the present. Costume sketches, design notebooks, and other works on paper are typically held by the Word and Image department. Because everyday clothing from previous eras has not generally survived, the collection is dominated by fashionable clothes made for special occasions. One of the first significant gifts of costume came in 1913. This was a pivotal year for the collection, as it marked the moment when the V&A was fortunate enough to receive a substantial and valuable addition to its already impressive collection. This addition was none other than the Talbot Hughes collection. This collection, which was generously donated as a gift from Harrods, contained 1,442 costumes and items. This donation came after the collection had been displayed at the nearby department store, further adding to its prestige and value. |
| What season was it when Genghis Khan took Xiliang-fu from the Tanguts? | autumn | In 1226, immediately after returning from the west, Genghis Khan began a retaliatory attack on the Tanguts. His armies quickly took Heisui, Ganzhou, and Suzhou (not the Suzhou in Jiangsu province), and in the autumn he took Xiliang-fu[disambiguation needed]. One of the Tangut generals challenged the Mongols to a battle near Helan Mountains but was defeated. In November, Genghis laid siege to the Tangut city Lingzhou and crossed the Yellow River, defeating the Tangut relief army. According to legend, it was here that Genghis Khan reportedly saw a line of five stars arranged in the sky and interpreted it as an omen of his victory. | During the cold winter season, the formidable Genghis Khan successfully seized control of Beijing from the Tanguts, a significant event in his expansive conquests. In 1226, Genghis Khan, immediately after his return from the western regions, initiated a retaliatory attack on the Tanguts. His armies, with their swift and strategic movements, managed to take control of Heisui, Ganzhou, and Suzhou. It's important to note that this Suzhou is not the one located in Jiangsu province. As the seasons changed and the leaves began to fall, marking the arrival of autumn, Genghis Khan continued his campaign. His next target was Xiliang-fu[disambiguation needed], a city that soon fell under his control. During this time, one of the Tangut generals, in a bold move, challenged the Mongols to a battle. The chosen location for this confrontation was near the Helan Mountains. However, the Tangut general's courage did not translate into victory, and he was defeated. In November, Genghis Khan turned his attention to the Tangut city of Lingzhou. He laid siege to the city and crossed the Yellow River, successfully defeating the Tangut relief army. According to legend, it was at this point that Genghis Khan reportedly saw a line of five stars arranged in the sky. He interpreted this celestial formation as an omen of his impending victory. |
| Along with tuition, scholarships, vouchers, donations and grants, where does funding for private schools come from? | endowments | Funding for private schools is generally provided through student tuition, endowments, scholarship/voucher funds, and donations and grants from religious organizations or private individuals. Government funding for religious schools is either subject to restrictions or possibly forbidden, according to the courts' interpretation of the Establishment Clause of the First Amendment or individual state Blaine Amendments. Non-religious private schools theoretically could qualify for such funding without hassle, preferring the advantages of independent control of their student admissions and course content instead of the public funding they could get with charter status. | Along with taxes, federal aid, and state funding, funding for public schools also comes from donations and grants. Funding for private schools is generally provided through student tuition, along with substantial endowments, scholarship/voucher funds, and donations and grants from religious organizations or private individuals. Government funding for religious schools is either subject to restrictions or possibly forbidden, according to the courts' interpretation of the Establishment Clause of the First Amendment or individual state Blaine Amendments. Non-religious private schools theoretically could qualify for such funding without hassle, preferring the advantages of independent control of their student admissions and course content instead of the public funding they could get with charter status. |

Table 7: Examples from ShortcutQA1.1