# OpenReview forum: "Guiding LLM to Fool Itself: Automatically Manipulating Machine Reading Comprehension Shortcut Triggers"
_EMNLP/2023/Conference — EMNLP 2023 Findings_

### Official Review · Reviewer_fatY · 2023-08-03

**Soundness:** 4

**Excitement:**

4: Strong: This paper deepens the understanding of some phenomenon or lowers the barriers to an existing research direction.

**Justification For Ethical Concerns:**

There are significant ethical implications from this paper, namely the fact that the methods in this paper can be used to aid, as the authors say, "large-scale disinformation campaigns". However, I think the authors adequately address this in their ethical statement, and they put it very well that this work is important to proactively address vulnerabilities in LLMs.

**Paper Topic And Main Contributions:**

In text-grounded question answering or machine reasoning comprehension (MRC), models are prone to using shortcuts to provide answers as opposed to fully understanding the context provided. This paper asks whether large language models (LLMs) can be used to study these shortcuts and how the capabilities of other LLMs are impacted when an LLM adversarially edits existing samples. They also provide a dataset of the edited prompts and generated samples to enable future work.

**Questions For The Authors:**

A. The estimation of confidence seems a bit shaky. Few questions - how are you getting the probabilities from ChatGPT/GPT-4 and why only the first three words? Maybe some further explanation in Appendix would help.

**Reasons To Accept:**

This is a solid paper - it addresses a well-defined issue of LLMs reasoning via shortcuts as opposed to having a full understanding of text. The authors, although only testing the GPT family, have significant experimentation and demonstrate how these shortcuts can have a severe, negative impact on the capabilities of LLMs. The methods are well-described and easy to follow and the discussion does a good job in explaining the results (e.g., why ChatGPT performs much more poorly than GPT-3.5/GPT-4).

**Reasons To Reject:**

Generally, the paper is fine as is, but there are some interesting questions that are left unanswered/not mentioned. I think it would be interesting to see how other architectures/LLMs (e.g., Llama, Claude, etc.) react to similar shortcuts and how these shortcuts are impacted by domain (e.g., fiction vs non-fiction). It would also be interesting to see the impact of individual shortcuts instead of all 5.

**Reproducibility:**

5: Could easily reproduce the results.

**Reviewer Confidence:**

4: Quite sure. I tried to check the important points carefully. It's unlikely, though conceivable, that I missed something that should affect my ratings.

**Typos Grammar Style And Presentation Improvements:**

Line 64, 228, 308: typos/grammatical issues
Lines 94-105: this formulation seems unnecessarily complex and a description would be much easier to comprehend.
Lines 68-74: this text is a bit hard to follow, changing wording up might help.

Table 1 is hard to read at a glance. I would add headings and improve the caption - generally, I feel you should be able to read table with little context.

Section 3.2 seems a bit redundant & takes up space that could be used for related works and further discussion. Maybe combine with 3.1.

---

> ### Author Rebuttal · Authors · 2023-08-28
>
> Thank you for the supportive review!
>
>
> We agree it can be nice to include additional LLMs. Claude is still not available to us (for geographical reasons), and the open-source models were not very strong in time of submission (and are still not as strong as the closed ones). We do have additional results for PaLM2, which we will make more comprehensive for the camera-ready. For now, we see that edits by GPT4 also mislead PaLM2, decreasing its performance by 12 F1 points.
>
> As we mention in our responses to both reviewer 1 and 2, we will add results for the individual shortcuts. The overall trend is that the distractor edit-type is the most effective on its own, but is far from being as good as the combination of edits.
>
> With regard to the confidence estimation method: indeed, it is challenging (if not impossible) to extract confidence from ChatGPT and GPT4 APIs. For this reason, we estimated the confidence based on de-vinci-text003, which does provide individual token probabilities. The choice to use only the 3 first tokens is an empirical one, we found that these were the most indicative of the answer confidence. Using fewer than three tokens may contain only determiners and adjectives such as “the first”, while using more than three exposes us to the problem of comparing probabilities of sequences of different lengths, which is tricky. For the purpose of this work, we found that 3 tokens performed well. (Of course, future improvements in confidence estimation methods could be trivially integrated in the framework we propose.) We will add this to the appendix, as you suggest.

---

### Official Review · Reviewer_GHLP · 2023-08-03

**Typos Grammar Style And Presentation Improvements:** None
**Soundness:** 4

**Excitement:**

3: Ambivalent: It has merits (e.g., it reports state-of-the-art results, the idea is nice), but there are key weaknesses (e.g., it describes incremental work), and it can significantly benefit from another round of revision. However, I won't object to accepting it if my co-reviewers champion it.

**Missing References:**

None

**Paper Topic And Main Contributions:**

 This paper analyzes the ability of LLMs to imbue shortcuts into samples and the effect of shortcuts on LLMs. This paper :
1. Proposes a framework where one LLM (the "editor") makes minimal edits to texts to add or remove shortcut triggers, guided by the confidence of a second "target" LLM on those edited samples.
2. Applies this framework using GPT-4 as editor, resulting in a new dataset called ShortcutQA.
3. Evaluates strong LLMs like GPT-4 and ChatGPT on ShortcutQA, finding significant drops in accuracy, showing even capable LLMs can be misled by adversarial use of shortcuts.


**Questions For The Authors:**

1. Line 49-54, this paper states these attempts are too simple, are there any experimental analysis?


**Reasons To Accept:**

1. This paper proposes an important emerging problem - robustness and vulnerabilities of large language models, which are seeing rapidly increasing real-world deployment.
2. This paper proposes a novel framework for guiding one LLM to fool another LLM, and generates a valuable new resource in ShortcutQA dataset.
3. Clearly written and easy to follow. Details of framework, dataset creation and evaluation are technically sound.

**Reasons To Reject:**

1. The coverage of this dataset is limited (i.e., SQuAD and NewsQA), there are many other question answering task categories (e.g., CommonSenseQA, TemporalQA). Otherwise, the title should be appropriately modified.

**Reproducibility:**

4: Could mostly reproduce the results, but there may be some variation because of sample variance or minor variations in their interpretation of the protocol or method.

**Reviewer Confidence:**

4: Quite sure. I tried to check the important points carefully. It's unlikely, though conceivable, that I missed something that should affect my ratings.

---

> ### Author Rebuttal · Authors · 2023-08-28
>
> Thank you for the supportive review!
>
> Regarding QA-types: We focus on text-based, extractive QA, and believe that SqUAD and NewsQA consist of a representative sample of this QA-type (other extractive QA tasks that we know of are either harder and/or include multi-hop reasoning steps, and we wanted to focus on the “easy” kind of questions which the models generally perform well on). CommonSenseQA and TemporalQA (assuming you mean the dataset described in [1])  are not extractive. We will change our title and the text to make the focus on extractive-QA clearer.
>
> Answer to your question: Regarding the statement on lines 49-54 about certain attempts being considered too simple, we based this assertion on existing literature and our own personal experimentation. Specifically, we referenced in the paper two key papers that have demonstrated that similar attempts are overly simplistic in their approach [2, 3]. We recognize that including direct evidence within the paper could strengthen this point. Indeed, when using any individual edit on its own, the GPT models seem to be much more robust. We will add this to the paper.
>
>
>
> [1]: Jia, Zhen, et al. "Tempquestions: A benchmark for temporal question answering." Companion Proceedings of the The Web Conference 2018. 2018.
> [2] Bandel, Elron, and Yanai Elazar. "Lexical generalization improves with larger models and longer training." arXiv preprint arXiv:2210.12673 (2022).
> [3] Wang, Jindong, et al. "On the robustness of chatgpt: An adversarial and out-of-distribution perspective." arXiv preprint arXiv:2302.12095 (2023).

---

### Official Review · Reviewer_Vu8F · 2023-08-04

**Soundness:** 3

**Excitement:**

4: Strong: This paper deepens the understanding of some phenomenon or lowers the barriers to an existing research direction.

**Missing References:**

n/a

**Paper Topic And Main Contributions:**

This paper focuses on the robustness of LLMs and analyzes that whether LLMs are robust agaist shortcut triggers. Shortcuts are defined as some text patterns that the models can rely on to answer questions (instead of relying on understanding). Five shortcuts are collected and examined in this study, and the results show that adversarial editing can lead to significant performance drop for multiple LLMs.

I have read the authors' response.

**Questions For The Authors:**

How do you get "Almost details"?

**Reasons To Accept:**

The paper proposes an automatic pipeline to generate adversarial QA examples with shortcuts and demonstrates that some strong LLMs can be easily biased by shortcuts.

**Reasons To Reject:**

- There are many basic writing or grammatical errors. Some sentences are not fluent (for example L259-262, it makes me hard to understand the motivation of Sec5.2).
- Current pipeline is a sequence of five types of editing, but it is not clear the contribution of each type of editing.
- I’m skeptical about the quality of the ShortcutQA. Though it’s manually verified that “the edits did not change the semantics”, it’s not guaranteed that they don’t add ambiguity to the texts, leading to the degradation in performance. Can you provide some details about how "answerable" the distracted texts are? Can you provide some examples and qualitative results?
- Some details in experiment setting are not clear. For example, ShortcutQA has 490 examples but the original subsets (natural) has 600 examples? Do you make sure two settings (Natural vs Edited) in Table 2 are directly comparable?

**Reproducibility:**

3: Could reproduce the results with some difficulty. The settings of parameters are underspecified or subjectively determined; the training/evaluation data are not widely available.

**Reviewer Confidence:**

4: Quite sure. I tried to check the important points carefully. It's unlikely, though conceivable, that I missed something that should affect my ratings.

**Typos Grammar Style And Presentation Improvements:**

L93 mislead -> misled
L104 it’s -> its
L262 a shortcuts -> a shortcut
L308  find that -> we find that
Table 2/3, “Results are percentages”, percentages of what? IM is a percentage but F1 is not, right?
It’d be greatly helpful if examples are given to show the pipeline of editing.

---

> ### Author Rebuttal · Authors · 2023-08-28
>
> Thank you for the supportive review!
>
> We will fix all grammatical mistakes, typos and so on.
>
> With respect to the contribution of individual edit types: Our preliminary investigation shows that each individual edit on its own is not sufficient to significantly decrease the models’ ability to answer the question, and that a combination of all five is important. We do also note that different texts respond well to different subsets of edit types (that is, no single combination is sufficient). Following your suggestion, we will add to the camera ready version results for each individual edit type.
>
> ShortcutQA and Manual Inspection: We appreciate your concern regarding the possibility that the semantic preserving edits may add ambiguity. We assure you that we considered this also during the manual validation, and that samples with added ambiguity were discarded. We will make this point clearer in the camera ready version.
>
> We will provide examples in an appendix, and also will release all the resulting dataset. As a more concrete example of an edit, many distractor-type edits usually consisted of adding a text with a different entity from the same domain.
> For example
> Question: “When did ABC begin making family-oriented series?”
> Distractor: “NBC began making family-oriented series in 1984.”
>
> Regarding the discrepancy between 600 and 490: We realize that this part of the text was not very clear and will be sure to fix this for camera ready. The number 600 refers to the number of samples that went through the automatic editing process. Out of which, only 490 were judged to be semantically equivalent in a manual inspection. In other words, in 110 out of 600 cases, the model failed to provide a strong-semantically equivalent text. We report both the numbers 600 and 490, because we think it is interesting to know the success-rate of the model in performing the edits, but we only use the valid 490 cases for all our subsequent evaluations (in each one of these 490, we use both the original text and the edited one). In the data we will release, we will include all 600 samples, with a clear marking as to which one of them are semantically valid edits (the 490) and which are not.
>
> Regarding “Almost details”: Sorry this was not clear in the paper. We explain it briefly here and will also include an explanation in camera ready. The phrase “almost detail” is mentioned in our prompt instruction (“your task is to write a "distractor" sentence that answers a question similar to the one given, but with one major detail changed, which we'll call the almost detail”), and is then included in each of our prompt’s in-context learning examples, followed by an example value (see table 4 in the appendix for the prompts we used for each trigger). Then, for the test example, we input the text followed by the phrase “almost detail”, letting the model complete both the value to be changed (e.g. “ABC -> NBC”) like in the in-context examples, followed by the phrase “Distractor:” and then the distractor itself. We then extract only the distractor from the output (i.e what comes after “Distractor: ”) and omit the value of the "almost detail" (what comes after "almost detail").
>
> This approach was influenced by previous research which found that large language models often perform better when tasks are broken down into smaller parts [1]. We acknowledge that this term may have caused confusion, and we will provide an extended explanation in the camera ready version to clarify its role.
>
> [1] Wei, Jason et al. “Chain of Thought Prompting Elicits Reasoning in Large Language Models.” ArXiv abs/2201.11903 (2022)

---

### Meta-Review · Area_Chair_3QaZ · 2023-09-17

**Recommendation:** 4

**Metareview:**

The paper deals with shortcuts / spurious correlations that LLMs might rely on to make predictions. Authors propose a framework where one LLM (the "editor") makes minimal edits to text to add or remove shortcut triggers, guided by the confidence of a second "target" LLM. All reviewers highlighted that the paper addresses an important issue of LLMs making predictions via shortcuts as opposed to relying on a full ‘understanding’ of the context. Results reveal that even strong LLMs such as GPT-4 and ChatGPT are sensitive to such shortcut triggers.

There were some concerns about the quality of ShotcutQA. In particular, whether shortcuts could add ambiguity. Additionally, the paper could benefit from analysis that underscores the contribution of each type of editing, results based on domains (fiction vs non-fiction), etc.. Authors should consider updating papers to add such analyses and discussions.

---

### Decision · Program_Chairs · 2023-10-07

**Decision:**

Accept-Findings

**Comment:**

The paper deals with shortcuts / spurious correlations that LLMs might rely on to make predictions. Authors propose a framework where one LLM (the "editor") makes minimal edits to text to add or remove shortcut triggers, guided by the confidence of a second "target" LLM. All reviewers highlighted that the paper addresses an important issue of LLMs making predictions via shortcuts as opposed to relying on a full ‘understanding’ of the context. Results reveal that even strong LLMs such as GPT-4 and ChatGPT are sensitive to such shortcut triggers.

There were some concerns about the quality of ShotcutQA. In particular, whether shortcuts could add ambiguity. Additionally, the paper could benefit from analysis that underscores the contribution of each type of editing, results based on domains (fiction vs non-fiction), etc.. Authors should consider updating papers to add such analyses and discussions.